# Shallow subsurface geology and seismic microzonation in a deep continental basin. The Avezzano Town, Fucino basin (central Italy)

Paolo Boncio<sup>1</sup>,Giuliano Milana<sup>2</sup>, Fabrizio Cara<sup>2</sup>, Giuseppe Di Giulio<sup>3</sup>, Deborah Di Naccio<sup>3</sup>, Daniela Famiani<sup>2</sup>, Francesca Liberi<sup>1</sup>, Fabrizio Galadini<sup>2</sup>, Gianluigi Rosatelli<sup>1</sup>, Maurizio Vassallo<sup>3</sup>

<sup>1</sup>Dep. DiSPUTer,"G. D'Annunzio" University of Chieti-Pescara,,Chieti, 66013, Italy
 <sup>2</sup>INGV, Rome, 00143, Italy
 <sup>3</sup>INGV, L'Aquila, 67100, Italy

Correspondence to: Paolo Boncio (pboncio@unich.it)

# 10 Abstract.

We present detailed geological investigations aimed at the reconstruction of the shallow subsurface geology, and associated local seismic hazard, of the Avezzano town in the Quaternary Fucino basin (central Apennines). This work shows basic (Level 1) seismic microzonation (SM) of the Avezzano town, focusing the attention on geologic constraints. We also discuss some methodological procedures of SM.

- Level 1 SM involves a reconstruction of the subsurface geological model achieved by a multidisciplinary approach synthesized in two main thematic maps and geologic sections. The first map, containing essential geologic information, is formed by overlapping layers (geological units, litho-technical units, and geomorphological/structural features). The second map is a summary map, easily accessible to non-geologist earthquake scientists/technicians, which synthesizes surface geology, subsurface data and resonance frequencies into homogeneous microzones. The two maps are tools for land and
- urban planning.

The Avezzano area provides a case study of shallow subsurface geology and site effects in a deep continental basin environment, and is of potential interest for similar geologic contexts worldwide. Within the investigated area, almost all the possible earthquake-induced effects can occur, such as a) stratigraphic amplifications in a wide range of resonance frequencies (from 0.4 to >10 Hz); b) liquefaction; c) coseismic surface faulting; d) basin-edge effects; and e) slope to the strategy of the strategy

instability.

#### Key words

Subsurface geological model, seismic microzonation, site effects, Central Italy, Avezzano.

## **1** Introduction

(i)

The surface and shallow-subsurface geology is extremely important for seismic site characterization. Geophysical investigations and their interpretation need to be anchored in a solid geological model. A geological model helps understanding the subsurface structural complexities, the 3D geometry of sedimentary bodies, their mutual relationships and

- 5 the relations with the bedrock interface. A geological model is achieved by the interplay of different data, but surface geology obtained by basic detailed geological survey (e.g., 1:5,000 scale), integrated with borehole stratigraphies, still remains the fundamental source of information. The synthesis of geologic data into graphic works (maps, sections, synthetic stratigraphies, etc) easily understandable from non-geologist earthquake scientists/technicians is a major challenge for seismic risk-oriented geologic works, such as seismic microzonation (SM). It involves an effort in synthesizing as many data
- as possible in as few elaborates as possible, reducing unnecessary details. 10 In this paper, we discuss the geological aspects, some methodological procedures of SM, and the implications on seismic site response of a urbanized area in a geologic environment of deep continental basin. In particular, we present the results of a basic SM (i.e., "Level 1" SM) of the Avezzano town, in the north-western corner of the Quaternary continental Fucino basin (Fig. 1). Avezzano was completely destroyed by the 1915 earthquake (M 7.0). The severity of the damage was certainly due
- to the proximity to the seismogenic source. In fact, the town is located in the hangingwall of the SW-dipping Fucino normal 15 fault system, activated during the 1915 earthquake. Nevertheless, the local geological conditions can have remarkably influenced the ground motion amplification. The reconstruction after the 1915 earthquake took place mostly along the marginal part of the basin, at the piedmont of the carbonate ridges bordering the basin. In more recent times, the urbanization expanded also towards the central part of the basin. The most depressed, flat area of the basin is devoted to agricultural 20 activity and ordinary urbanization is not allowed.
- The SM was carried out according to the guidelines for SM by the Italian Department of Civil Protection (Working Group SM 2008), implemented with additional methodological procedures shared with the Abruzzo Regional Authority (http://protezionecivile.regione.abruzzo.it/index.php/microzonazione).

## 2 Geological setting

25 The Fucino basin is a Quaternary graben located in the core of the central Apennines of Italy. The graben hosted a large lake, which was drained at the end of the XIX century. It is one of the several Quaternary intermontane extensional basins of the Italian Apennines. The basin is superposed on a Neogene fold-and-thrust belt constituted by Mesozoic to Tertiary passive margin carbonates and syn-orogenic flysch deposits.

The evolution of the Fucino basin is related to the activity of two main fault systems (Fig. 1b). The first system strikes WSW-ENE and dips to the SSE (Tre Monti normal fault system); the second system strikes NW-SE and dips to the SW 30 (Fucino normal fault system).

Though the SSE-dipping Tre Monti system had an important role during the early evolution of the basin (Late Pliocene-Early Pleistocene), the master fault of the graben is the SW-dipping Fucino fault system (Galadini and Messina 1994), which borders the basin towards the east. Both the fault systems have normal kinematics, with left-lateral strike slip component observed for the NW-SE fault system at some intervals during the Quaternary (Galadini and Messina 1994). Seismic

- reflection and well data show a typical half-graben sedimentary infill in the hanging wall of the Fucino fault system (Cavinato et al. 2002; Patacca et al. 2008) (Fig. 1b). In fact, the basin depocenter is localized towards the eastern margin, with estimated thickness of the basin infill of about 1000 m (Cavinato et al. 2002). The Quaternary continental deposits unconformably cover a bedrock formed by Mesozoic-to-Middle Miocene carbonates, cropping out in the reliefs surrounding the basin (Mt. Salviano and Tre Monti ridges westward and northward, and Mt. Parasano- Mt. Serrone ridge eastward), and
- by Late Miocene siliciclastic turbidites, mostly buried under the Quaternary sediments but largely exposed westward, along the Val Roveto valley. Concerning the continental infill, in the central part of the basin the stratigraphy is dominated by Holocene fine-grained lacustrine sediments (silt and clay, Fig.1b). Along the perimeter of the basin, Pleistocene medium-to-coarse-grained fluvial, alluvial fan, and slope-derived deposits are interfingered with lacustrine sediments (Fig. 1b). The perimeter area is also characterised by a complex morphology, with several erosional and depositional surfaces (Giraudi 15, 1089).
- 1988).

On the base of field and subsurface data, Cavinato et al. (2002) relate the early stage of the basin evolution and the deposition of the Lower Pleistocene sediments to the activity of the WSW-ENE Tre Monti Fault system. While the NW-SE-striking Fucino fault system controlled the deposition of the Upper Pleistocene-Holocene sediments.

The Fucino basin is a seismically active area, as highlighted by the occurrence of the January 13, 1915 earthquake (Ms=7.0, MCS Intensity XI) (CPTI15; Rovida et al. 2015), after which a low-seismicity period occurred (Fig. 1a). Paleoseismological studies carried out on the Fucino fault system (Michetti et al. 1996; Galadini and Galli 1999) reveal the occurrence of at least 10 paleoearthquakes in the last 33 kyrs, with a return period ranging from 1400 and 2600 yrs (Galadini and Galli 1999).

# 3 Methodology

Seismic microzonation(SM) is a tool used to evaluate the local seismic hazard of an area exposed to earthquakes and aims to subdivide it into zones having homogeneous seismic site response.

Following the Italian guidelines for SM ("Indirizzi e Criteri per la MicrozonazioneSismica"; Working Group SM 2008), SM studies are organized in three different levels of knowledge. Level 1 SM is an indispensable step, and is preparatory of the levels 2 and 3, because all the available geological data and constraints are collected in this first phase. Level 1 SM involves a reconstruction of the shallow subsurface geology, achieved by using detailed field mapping, pre-existing subsurface data

and new geophysical data collected basically from recordings of ambient seismic noise. The main outputs for Level 1 SM are thematic maps and geologic sections that highlight the possibility of ground-motion amplifications due to stratigraphic and/or topographic conditions, liquefaction processes, slope instabilities and coseismic deformation due to active and

capable faults. During the following Levels 2 and 3 SM, key areas are investigated in details with the acquisition of new data through ad-hoc experiments and by quantitative site-response analyses carried out by simplified approaches (Level 2) or numerical methodologies (Level 3).

This paper focuses on the Level 1 SM of Avezzano, which was organized in 4 phases:

1) Collection, quality-selection and georeferencing of all the pre-existing geological, geognosticand geophysical investigations:

2) geologic and geomorphologic field survey at 1:5,000-scale, mapping and construction of detailed geological sections;

3) Single-station recording of ambient seismic noise (65 new recordings) and analysis with the Horizontal-to-Vertical Spectral Ratio technique (HVSR), in order to enhance site effects of the area (Nakamura 1989, 2000). In fact, in case of

- simple geological model and high impedance contrast between sedimentary filling and stiff bedrock, the peak of H/V noise spectral ratios corresponds to the resonance frequency of soil (e.g. Field and Jacob 1993; Duval et al. 1994, 1995, Bonnefoy-Claudet et al. 2006). The analysis allows to define: i) the zones where the H/V curve is flat and likely unaffected by amplification of the ground motion; ii) the zones where the H/V curve shows a peak and therefore characterized by site amplification; and iii) the likely resonance frequency  $(f_0)$  for the amplifying sites. The relationships between  $f_0$  and the local
- stratigraphy haveimportant implications on how HVSR data can be used to infer the depth of the seismic bedrock in areas where other geological-geophysical data are lacking. Details about the used instrumentation and data processing, and a comparison with weak motion from earthquakes can be found in Famiani et al. (2015; see also Cara et al., 2011); 4) Synthesis of the data in a Map of Homogeneous Microzones from a Seismic response Perspective (MOPS in the Italian guidelines for SM; Working Group SM 2008).
- Point 2 is the most important during Level 1 SM as the knowledge of the local subsurface geological setting is necessary for 20 a seismic response analysis. Adetailed geological survey has been carried out both on the urban zones and the surrounding area planned to be urbanized within the Avezzano municipality (Fig. 4). This phase is finalized to produce a geological map (1:5,000 scale) containing several technical features useful for SM such as lithology, physical-mechanical properties of soils (texture, compactness/strength), structural (e.g. fracturing of rocks), hydrogeological and geomorphologic features relevant
- for seismic site response (Fig. 2; see Fig. S1 of the auxiliary material for a 1:5,000-scale map of the Avezzano town area). This map, called here Geological-Technical Map for SM, is the final product of phase 2 and comes from the superposition of different data layers stacked on top of each other. Thestructure of the Geological-Technical Map presented in this paper is new compared to existing guidelines (e.g., Working Group SM 2008), and is aimed at preserving basic geologic information together with technical data important for local seismic hazard. In particular, the map is formed by three main layers (Fig
- 2a:

•Layer 1 represents the geological base map and contains the geological-stratigraphical features. The classification of the pre-Quaternary geological bedrock refers to the 1:50,000-scale Italian Geological Map of the CARG project, sheet n°368 "Avezzano" (available at http://www.isprambiente.gov.it/Media/carg/368\_AVEZZANO/Foglio.html).

5

•Layer 2 contains the litho-technical units (Fig. 2b), which provide a physical-mechanical classification of soils (cover units) and bedrock. The using of litho-tecnichal units follows the methodological procedures adopted by the Tuscany (northern Italy) Regional Authoritysince '90 (e.g., L.R. 30.7.97 n.56 "Programma VEL ValutazionedegliEffettiLocali -Istruzionitecniche per le indaginigeologiche, geofisiche, geognostiche e geotecniche per la valutazionedeglieffettilocalineicomuniclassificatisismicidella Toscana", available online at www.regione.toscana.it/documents/10180/12216795/volume6.pdf and formalized with internal guidelines of the Abruzzo

Regional Authority available at  $http://protezionecivile.regione.abruzzo.it/files/rischio\%20 is smico/microzonazione/OPCM3907/LineeGuidaMS\_v1\_2\_ONLI$ NE2.pdf). The Bedrock is classified into 4 main units, including massive rocks (A), stratified rocks (B, with 6 sub-classes to

- account for bed thickness and presence of pelitic interlayers with lower rigidity), granular cemented bedrock (C, with 3 sub-10 classes to account for grain size and grain- or mud-supported texture) and over-consolidated pelitic bedrock (D, with 2 subclasses to account for grain size). Cover units are distinguished into two classes and numerous subclasses to account for grain size and texture. They include coarse granular uncemented or poorly cemented soils (E, with 7 subclasses to account for mixtures between gravel and sand) and fine or organic soils (F, with 5 subclasses to account for organic soils and
- 15 mixtures between silt and clay). Roman numerals are used to classify the different compactness/strength of coarse/fine soils (Fig. 2b).

As an example, six litho-tecnichal units are mapped in Fig. 2c. The geological bedrock is classified asB1 andgroups medium-to-very thickly bedded rock masses. Cover units are classified as E (uncemented, coarse granular soils with different grain size and compactness), and F (fine soils) units. Anthropic and waste materials are incorporated inunit G.

- •Layer 3 contains structural, geomorphological and some hydrogeological elements, focusing the attention on those aspects 20 significant for microzonation purposes (active and capable faults; buried faults that may determine bedrock steps; shallow aquifers; topographic features such as peaks, crests, morphological scarp; slope instabilities). Concerning the definition of "active and capable fault", the term "active" indicates a fault having evidence of repeated activations during the late Quaternary (in particular, during the last 40 kyrs according to Working Group SM 2008), and the term "capable" indicates
- the capability to rupture up to the surface during large earthquakes, therefore determining a local seismic hazard (surface 25 fault rupture hazard).

# 4 Pre-existing subsurface data

The shallow subsurface geology of the soft cover units, the depth of the geologic bedrock and the identification of buried structures of the bedrock are prerequisites to evaluate areas potentially susceptible to site effects. With this purpose our 30 analysis starts with the collections of all the available subsurface data, including geophysical, geognostic and geotechnical

investigations, hydrogeology and groundwater data, and subsurface stratigraphy from well data. The quality of the data are then validated and organized into a geodatabase (Fig. 3).

Overall, we collected 435 data which mainly consist of: stratigraphies of water wells (Italian National Law n. 464/1984, made available by ISPRA, Italian Institute for Environmental Protection and Research; and old wells drilled during '50 by

- 5 "EnteFucino", a local managing institution), geophysical investigation for hydrological exploration (Vertical Electric Sounding,VES), several stratigraphies and mechanical log characterizations from geognostic wells made available by professional geologists. In particular, 35 wells of the 160 collected (geognostic or for water exploration/exploitation) were drilled up to the geological bedrock (maximum well depth 270 m). Moreover, for the Castello Orsini site within the Avezzano town (Fig. 3) we collected HVSR data in 2D seismic array configuration used to characterize the AVZ strong-
- 10 motion station of the Italian Network (ITACA working group 2016) and Seismic Dilatometer Tests (SDMT) by University of L'Aquila. Industrial seismic-reflection profiles available for the entire Fucino area (Cavinato et al. 2002) and a deep reflection seismic profile (CROP 11 line, Patacca et al. 2008) provide information on both the deep structure of the continental basin and the depth of the pre-Quaternary bedrock (Fig. 1b).

# 5 Surface and shallow subsurface geology

#### 15 5.1 Surface geology

The surface geology is synthesized in the geological map of Fig. 4. The pre-Quaternary bedrock, cropping out in the Mt. Salviano and Tre Monti ridges, is formed by Cretaceous neritic limestones passing upwards to Miocene limestones, directly or through a more continuous succession of marginal-slope deposits of Cretaceous-Paleogene age. The carbonate successions are covered by pre-flysch (marly limestones, marls and marlypelites) and pelitic-arenaceous flysch deposits of

Late Miocene age. These units correspond to the pre-Quaternary geological units reported both in the Geological Map of Fig.
 4 and in the Geological-Technical Map of Fig. 2.

The continental Quaternary deposits unconformably overlay the bedrock units and are formed by slope, alluvial fan and fluvial medium-to-coarse-grained deposits that are interfingered with fine-grained lacustrine deposits towards the central part of the basin. In the area of Fig. 4, the maximum thickness of the Quaternary deposits is thought to be about 350 m (northern

25 and eastern sectors).

Numerous works have been published in the last twenty years on the stratigraphy of the Quaternary continental deposits (Zarlenga 1987; Giraudi 1988; Galadini and Messina 1994; Bosi et al. 1995; Cavinato et al. 2002; Centamore et al. 2006). More recently, three main stratigraphic domains have been defined in the CARG geological map (http://www.isprambiente.gov.it/Media/carg/368\_AVEZZANO/Foglio.html):

30 1) The first domain includes old fluvial and lacustrine deposits, with thick interlayers of slope-derived massive breccia, which crop out in the northern boundary of the basin (Lac1 and Ver1 in Fig. 4; Lower-Middle Pleistocene "Aielli-Pescina"

supersynthem and Middle Pleistocene "Catignano" synthem in the CARG geologic map). Their origin is related to the former opening of the basin, and they are often faulted and uplifted in the footwall of the main normal faults.

2) The second domain includes the marginal area of the lacustrine depression, where fine-grained lacustrine sediments (silt and clay) are interlayered with coarse-grained (sand and gravel) alluvial, deltaic and shoreline deposits (Lac2 and All2 in 5 Fig. 4; Upper Pleistocene "Valle Majelama" synthem in the CARG geologic map);

3) The third domain includes the central part of the basin, where the stratigraphy is dominated by fine-grained lacustrine sediments (silt and clay), with an increasing percentage of sand in the areas close to the margins (Lac3 in Fig. 4). This area was occupied by the Fucinolake, which was completely drained by the end of the XIX century. The age of the first few meters is Late Pleistocene (upper part) - Holocene (Giraudi 1988).

In this study, the Quaternary geological units (Fig. 4) were distinguished on the base of sedimentary environment and age as follow:

The lacustrine deposits re divided into Lac1, Lac2 and Lac3 units (Fig. 4).

"Lac3" characterizes the third stratigraphic domain and is formed by prevailing silt and clay with sand, peat and pyroclastic interbeds.

"Lac2"(second domain) is made up of rounded gravels with scarce or no matrix, typical of lacustrine shoreline, alternating sandy gravel and gravelly sand of lacustrine deltaic facies and silty sands. They deposited in the marginal area, where many erosional surfaces are present due to oscillationsof the lake level (Giraudi 1988).

"Lac1" (first domain) is formed by alternating silts and clayey silts with minor interlayers of micaceous sands.

The alluvial deposits are divided into All2 and All3 units, which can be correlated with Lac2 and Lac3 units, respectively.

- "All3" is formed by rounded to sub-rounded calcareous gravels, loose or slightly cemented sands and silty sands constituting the San Pelino alluvial fan and other recent, minor alluvial fans located at the outlet of small valleys (Fig. 4).
  "All2" is formed by rounded to sub-rounded calcareous gravels and sandy gravels, occasionally stratified. All2 deposits
  - correspond to two large alluvial fans that fed the lacustrine basin during Late Pleistocene cold climatic conditions (Valle Majelama and Celano alluvial fans; Frezzotti and Giraudi 1992).
- The slope deposits are divided into Ver1 and Ver2 units (Fig. 4). "Ver2"crops out at the base of carbonaceous reliefs and is formed by medium- to coarse-grained bodies of loose to dense slope-derived calcareous gravels in a sandy-silty matrix. The matrix is brown to dark-red and vary from very abundant (matrix-supported) to absent. The age is Late Pleistocene-Holocene.
- "Ver1" is formed by ancient massive or coarsely stratified breccias. We recognized three lithotypes: i) well-cemented calcareous breccia, with angular, heterometric clasts in sandy matrix; ii) stratified calcareous breccia with red, silty matrix; iii) less cemented, whitish calcareous breccia without matrix. The age is Early-Middle Pleistocene.

"Coll" is a unit formed by colluvial-eluvial deposits accumulated along the foothills, heterogeneous both in grain size and textures. The age is Late Pleistocene-Holocene.

"Ant" and "dis" are local covers made up of backfill/anthropic material and waste material, respectively.

The structural elements are mainly represented by WSW-ENE-and NW-SE-striking normal faults, related to the fault systems bounding the Fucino basin (e.g. the Tre Monti and Fucino fault systems). Some are buried faults, inferred from well and seismic reflection data (e.g., seismic reflection data from Cavinato et al. 2002). These faults have no evidence of late Quaternary activity and are responsible for the formation of bedrock steps. Others faults display clear morphological and/or

- paleoseismological evidence of late Quaternary activity and are considered active and capable faults. This is the case of Tre Monti, Trasacco and LucodeiMarsi faults. The NW-SE-striking Trasacco and LucodeiMarsi faults are, respectively, synthetic and antithetic splays of the Fucino main fault system. Paleoseismological studies testify repeated reactivations of the SW-dipping Trasacco fault during large earthquakes, the last one being the 1915 earthquake (Giraudi 1986, 1988; Galadini et al. 1997, 1999; Galadini and Galli 1999). The NE-dipping LucodeiMarsi fault, located by Giraudi (1986) close to
- the LucodeiMarsi village, has been uncertainly prolonged to the NW up to the industrial area of Avezzano (Fig. 4) on the base of paleoseismological trenching (Galadini et al. 1997, 1999; Galadini and Galli 1999) and well data analysed in this work (see section G-G' in Fig. 5). Paleoseismological data display at least two reactivation episodes along the LucodeiMarsi fault, both consistent with the seismic history of the Fucino main fault system. The Tre Monti fault has geological evidence of continuous tectonic activity during the entire Quaternary (Galadini and Messina 1994; Bosi et al. 1995; Piccardi et al.
- 1999; Galadini and Galli 1999; Morewood and Roberts 2000). The fault dips SSE with a 50-60° dip angle and a transtensional dextral kinematics. The ENE-WSW-striking Paterno fault is the basal segment of the Tre Monti fault system, bordering the Fucino lowlands to the north. It is not clear if the Paterno fault is presently active. In fact, subsurface data show that this fault was highly active
- during a long part of the Quaternary, as it was responsible for a sharp deepening of the bedrock in its hangingwall under the Lower-to-Upper Pleistocene lacustrine deposits (Lac1 and Lac2) (section A-A' in Fig. 6). On the other hand, there are no constraints on the late Quaternary activity of the fault. Only Giraudi (1988) documented discontinuous fault scarps along the Paterno fault trace, NE of the Paterno village, on Upper Pleistocene lacustrine shoreline deposits. Therefore, the Paterno fault (eastern segment) has been classified as "potentially active and capable fault". This means that only additional earthquake geology investigations, typical of higher level SM (i.e., Level 3 SM) might resolve the uncertainties.

# 25 5.2 Geological sections

30

The subsurface geology of the area is synthesized along several2D geological sections realized in key areas for the reconstruction of the subsurface geological model. The most significant sections are shown in Fig.s 5 and 6. The shallow subsurface geology of the Avezzano area can be synthetically divided in three typologies: A) Simple basin-edge geometry; B) Sharp basin termination due to boundary normal fault; and C) B-type geometry covered by a thick layer of fan gravels (zone north of Avezzano, covered by the large Valle Majelama alluvial fan).

In detail, the simple basin-edge geometry (A) characterizes the western edge of the Fucino basin (section G-G', Fig.5), with a progressive, gentle deepening of the bottom of continental infill towards the centre of the basin. Section G-G' is constrained by a number of deep geognostic wells, some of which drilled up to the carbonate bedrock. The western part of

the section is characterized by a nearly-horizontal surface on the carbonate bedrock. It is an erosional surface carved by the water of the Fucino Lake during the last glacial maximum, when the lake level reached its maximum height (~18-20 kyrsago; Giraudi 1988). East of the geognostic wells the depth of the bedrock is constrained by seismic reflection data (Cavinato et al. 2002). The continental infill shows a progressive thickening towards the centre of the basin. A small step of

- the top of the bedrock, due to the LucodeiMarsi normal fault, is inferred from well stratigraphies, but it does not modify significantly the simple geometry of the basin edge. The estimated maximum depth of the bedrock is ~200 m beneath point G'. It further deepens eastwards as shown by the isochrones of the top of bedrock (Fig. 1c, Cavinato et al. 2002). The time-to-depth conversion is usually problematic, due to lateral seismic wave velocity variations within the lacustrine deposits produced by a number of variables such as porosity, compaction and grain size variations. Considering that coarser-grained
- sediments increase towards the margin of the basin, it is likely that also the average seismic wave velocity increases accordingly. Nevertheless, due to the lack of firm and homogeneously distributed constraints, we performed a time-to-depth conversion by using an average P wave velocity (Vp) of 2000 m/s, independently from the position within the basin. This choice derives from an average of Vp values calculated in three sites where the isochrones by Cavinato et al. (2002) are compared to the depth of the bedrock from boreholes (two sites) or estimated from surface-wave analysis based on noise 15 data (AVZ station in the Castello Orsini site. ITACA working group 2016).
- data (AVZ station in the Castello Orsini site, ITACA working group 2016). The B-type margin is characterized by a sharp termination due to a boundary normal fault in the northern edge of the basin (section A-A' in Fig.6a). The A-A' section intersects the Tre Monti and Paterno normal faults, both located at the base of the Tre Monti ridge. The carbonate bedrock crops out in the footwall of the Tre Monti fault while the deposits belonging to Ver1 and Lac1 units are present between the two faults. In the footwall of the Paterno fault, close to the fault trace, the bedrock is
- likely to be formed by Miocene siliciclastic turbidites (UAP-B4 in Fig. 6), located at a depth of 20 m from the ground surface (Bertini and Bosi 1976). In the hanging wall of the Paterno fault the bottom of the Quaternary infill sharply deepens to ~150 m. The textural features of the continental deposits are doubtful as no direct geognostic investigations are present in this area. However, it is likely that the ancient lacustrine deposits (Lac1) are interfingered with coarse-grained continental deposits (Ver1), corresponding to the units that crop out in the footwall of the Paterno fault, or deriving from disruption and
- sedimentation of the footwall units. The Upper Pleistocene Lac2 deposits are only present in the hangingwall of the Paterno fault.

The C-type margin consists of a sharp termination geometry covered by a thick layer of gravels, shown in the geological sections C-C' and D-D' of Fig.s 6b and 6c. The gravels belong to the distal part of the Valle Majelama alluvial fan (All2, Fig. 4), which prograded significantly within the lacustrine basin during the last glaciation period of Late Pleistocene

(Frezzotti and Giraudi 1992). The gravel thickness and the transition to the underlying lacustrine units are constrained by a number of geognostic and water wells (Fig. 6). It is worth to note that the presence of stiff deposits (All) on soft deposits (Lac) and their thickness may cause local inversion of the seismic wave velocity with depth, which can be important in terms of seismic site response. The depth of the top of bedrock, formed mainly by Miocene siliciclastic turbidites (UAP-B4 in Fig. 6), is constrained by seismic reflection data (Cavinato et al. 2002) and by water wells in the north-westernmost part. The

presence of SE-dipping normal faults accounts for the progressive lowering of the bedrock units, with the Upper Pleistocene alluvial fan gravels sealing the faults.

## 6Geological bedrock, site response from microtremoranalysisand homogeneous seismic microzones

## 6.1Geological bedrock and resonance frequencies of the overlying cover units

- In the Fucino basin, carbonate and siliciclastic flysch geological bedrocks underlay the Quaternary fluvio-lacustrine infill. The pre-Quaternary bedrock crops out in the Mt. Salviano and Tre Monti ridges, where it is mainly formed by ordered thickto medium-bedded carbonate rocks and, subordinately, by alternating carbonate and pelitic rocks. Down-hole and cross-hole investigations collected in the Industrial area of Avezzano indicate Vs of ~1000 m/s close to the ground surface, increasing to > 2000 m/s at 10 m depth, for the carbonate bedrock (western side of section G-G' in Fig. 5). Direct near-surface
- measurements of Vs for siliciclastic flysch are not available, but the Vs profile obtained by passive 2D array in the center of the Avezzano suggest Vs on the order of 2000 m/s for siliciclastic flysch at depths of ~ 160 m (station AVZ in ITACA working group 2016). The same Vs profile indicates a mean Vs of ~200 m/s for the first 35 m of lacustrine sediments (Lac2 in Fig. 4), and a Vs of ~ 500 m/s for the thick pile of lacustrine sediments from 35 to 160 m depths (Lac1+Lac2 in sections of Fig.s5 and 6). Therefore, a strong impedance contrast is expected between geologic bedrock (carbonate or siliciclastic)
- and Quaternary continental infill. The impedance contrast might decrease appreciably where the Quaternary infill is dominated by dense, coarse-grained sediments, such as the alluvial fan deposits (northern side of the basin, sections A and C of Fig. 6). In fact, available down-hole investigations in the Valle Majelama alluvial fan indicate Vs of 540 - 710 m/s in the first 30 m depths.

Fracturing might decrease the Vs in rock masses. Therefore, a geomechanical field survey has been performed along the eastern side of the Mt. Salviano ridge in order to characterize the degree of fracturing of rock masses according to the International Society for Rock Mechanics (I.S.R.M. 2007) procedures. The volumetric joint count (Jv) measurements give values varying from 10 to 20 (except for one site where Jv ≥ 30) indicating that the geological bedrock is formed by moderately fractured rock masses.

For site response evaluations it is important to define the impedance contrast that can cause resonance amplification of

- seismic waves. The fundamental resonance frequencies obtained from HVSRs and their spatial distribution certainly help, as they depend on the lithology and Vs of the basin infill, the depth of the bedrock and the impedance contrast between the sedimentary cover and carbonate or flysch bedrock. Sites located over outcropping bedrock provide flat H/V curves, unless complications due to fracturing or topographic irregularities are present. Flat H/V curve (i.e., H/V ~ 1 over the 0.2 - 20 Hz interval) is a necessary, but not sufficient, condition to exclude local site amplification of the ground motion. In general, in
- the studied area the outcropping bedrock shows nearly flat H/V (e.g., Fig. 5). H/V peaks > 2, with fundamental resonance frequency ( $f_0$ ) generally lower than 1 Hz characterize sites located in the most central and deepest part of the basin while  $f_0$  higher than 1 Hz is obtained for stations installed in the marginal area of the basin.