# Peer review of "Shallow subsurface geology and seismic microzonation in a deep continental basin. The Avezzano Town, Fucino basin (central Italy)"

_Natural Hazards and Earth System Sciences, 2016_

## Referee Comment (RC1) · Anonymous Referee #1 · 14 Nov 2016

The paper presents the level 1 Seismic Microzonation of the Avezzano town considered as a case study of shallow subsurface and site effects in a deep continental basin environment, focusing on geologic constraints and discussing some methodological procedures.

GENERAL COMMENTS: Despite the topic is interesting and suitable for NHESS (subsurface geology and associated local seismic hazard in anthropized continental basin), the goals of the manuscript are not clearly presented; the authors seem to hesitate in identifying the most important and innovative results of their own work, describing data, methods and results (achieved or open questions) without emphasis (e.g. the proposed new structure of Geological-Technical Map for SM is wound up in few sen-

tences without highlighting the strengths and differences with existing guidelines). This is evident from the very beginning: i) the title refers to seismic microzonation in a deep continental basin but data, methods, maps, and results refer almost exclusively to the shallow part of the basin (the Avezzano town and surroundings); ii) the Introduction is too general and vague (e.g. the "additional methodological procedures" proposed as implementation of guidelines for Seismic Microzonation should be stressed and better described due to their relevant applicative implications); iii) in the further paragraphs new data and implementation of new methods are far from being clearly separated from literature or previous data and methods.

The paper can be improved by clearly defining the main goals and going through the rewriting of some paragraphs, first of all the Introduction, in order to clarify the perspective. My recommendation is that the manuscript can be accepted after revisions according to general and specific comments.

SPECIFIC COMMENTS TITLE The title should be modified to better fit with the content of the manuscript. The use of "deep" referred to the (Fucino) continental basin as a whole is misleading and in contrast with "The Avezzano Town", indeed the paper deals with the seismic microzonation of the Avezzano town, that is not the deeper part of the basin. More in general, considering the map of fig. 1b, the Fucino basin can't be considered as a deep continental basin due to the very limited extension of its deeper part (Quaternary infill up to 1,000 meters). I suggest to modify the title deleting the catchword "deep" and putting in evidence the case study of the Avezzano town. Please don't use the capital letter for Town.

1. INTRODUCTION - The section must be rephrased, especially in the first part, where the many clauses don't make a meaningful sentence. - The Authors state that "a geological model is achieved by the interplay of different data, but surface geology obtained by basic detailed geological survey (e.g. 1:5,000 scale), integrated with borehole stratigraphies, still remains the fundamental source of information"; however, considering the morphology of the Fucino basin (flat area) and that the paper focus on the Avez-

zano town (urbanized area) I suggest the Authors to better explain what they mean and how they have used surface data as fundamental source of information. Also in the paper the subsurface data seem to be the most important ones.

2. GEOLOGICAL SETTING - In the text (p. 2 line 25 and p.3 line 5) it is not clear if the Authors consider the Fucino basin as a graben or half-graben; they use (Fig. 1) the geological section from Cavinato et al., 2002 with a half-graben interpretation but the map where the trace of the cross section is indicated reported an articulated graben framework for the Fucino basin not corresponding with the cross section. Please better explain your personal interpretation, based on your own new data, and draw your own geological cross section. - the Authors use the term Late Pliocene (p.3 line 1): it would be useful to indicate the chronostratigraphic chart used or to modify the term with Gelasian in order to give a clear and unambiguous chronostratigraphic reference.

3. METHODOLOGY - The phase 2 "geologic and geomorphologic field survey" is indicated as the "most important during Level 1 SM" and the Authors state "A detailed geological survey has been carried out". However data and results of this activity are not well presented in the paper. As reported in the text, most of the surface geology characteristics are derived or referred to the 1:50,000-scale Italian Geological Map of the CARG Project (see p. 4 line 33, and pp.6-7). Please try to better explain which are your own new data. - I suggest to define with short titles the 4 phases and to use the same titles in sections 4-6. - At the web page linked at page 4 line 33 the geological sheet Avezzano is reported as published in 2005. Please add the correct reference to the map in the text (pp. 4, 6-7) and in the references list, together with the web link. - p. 4 line 22: Fig. 4 is cited before fig. 2 and 3. You can delete this reference, it is not necessary here. - p. 5 line 12 : cover units are distinguished into three classes (as in fig. 2a) and not in two as indicated in the text.

4 AND FOLLOWING SECTIONS I suggest to use the same titles adopted for the 4 phases described at page 4. Section 5.1 - p. 6 line 28: what do you mean with stratigraphic "domain"? Section 5.2. - p. 8 line 28-29, 31: use here the same style for

the "name" of the three typologies of shallow subsurface, as in the text at p. 9 lines 16 and 27, and in the figures. i.e.: "A-type....; B-type....; C-type......". - p. 9 line 11-14: the time-depth conversion used appears too simple for a basin characterized by high stratigraphic complexity and high lateral variability of the units. Do you have tested different values or more complex velocity models to have a more consistent prediction of the bedrock depth in the whole area? Section 6.3 p. 12 line 9: "contained in the MOPS" replace with "summarized in the MOPS"

FIGURES Fig. 1 - The isochrones from Cavinato et al., 2002 appear to be modified also outside the rectangle of Fig.4. Please check and/or modify the caption. - The geological cross section should be replaced with your own cross section or the trace moved in fig. 1a. As a matter of fact the trace on the map of fig. 1b creates misunderstanding in the reader due to the differences between faults in the map and in the geological cross section. Fig. 3: not clear/unreadable Fig. S1: please indicate that some of the units included in the legend are not represented in the map, or modify the legend. Fig. S2: please indicate that some of the zones included in the legend are not represented in the map, or modify the legend.

REFERENCES - Add reference to the geological sheet Avezzano - Check the correct form of the publication MS Working Group, 2008. In the text is always cited as Working Group SM, 2008. Unify. - SESAME 2004: it is not cited in the text.

TECHNICAL CORRECTIONS - there are several words not separated by blank - p. 13 line 31: "undergoe" change in "undergo"

---

## Referee Comment (RC2) · Anonymous Referee #2 · 24 Nov 2016

GENERAL COMMENTS

This paper has only applicative interest because it focuses on the Level 1 SM, which is a routine practice in Italy.

SPECIFIC COMMENTS

They write: "25 The Fucino basin is a Quaternary graben" but, strictly speaking, the section of figure 1b does not show a graben. Then, in Fig. 1b the trace of the section intersects 5 red faults, but the section below shows 6.

They write that "the SW-dipping Fucino fault system [...] borders the basin towards the east" and "the basin depocenter is localized towards the eastern margin". Rather, it

borders the basin towards the NE, and they refer to the north-eastern margin.

"Pleistocene medium-to coarse-grained fluvial, alluvial fan, and slope-derived deposits are interfingered with lacustrine sediments (Fig. 1b)". Non visible in the figure.

"from 1400 and 2600 yrs" from 1400 to 2600 yrs (?). However, 33 kyrs/9 = 3666 yrs. So, "from 1400 to 2600 yrs" does not work. Thus?

I think they used Lennartz 3D/5 sensors. Please mention the used instrumetal chain also in this paper, to ensure the reader that detected frequencies < 1Hz are reliable.

TECHNICAL CORRECTIONS

The Mt. Parasano and Mt. Serrone ridge and the Val Roveto are not shown in the figures.

Fig. 4 is mentioned before Figs. 2 and 3. Change their order.

"(silt and clay, Fig.1b)" are not shown in Fig. 1b.

Several cases of irregular spacing throughout the text.

The SDTM and the dynamic penetration test, shown in Fig. 3, are not mentioned in the text

---

## Referee Comment (RC3) · Anonymous Referee #3 · 28 Nov 2016

The manuscript describes the construction of a level 1 seismic microzonation map of the Avezzano town in central Italy. It begins with an overview of the studies area, proceeds with describing what a seismic microzonation is, and continuous with describing the types of data collected for this research, and the data itself.

General comments The work itself is very local, although it can serve as a "guide" for other places in the world. The authors state that the SM was carried out according to the guidelines for the SM by the Italian Department of Civil Protection. It is not understood whether there are new approaches or methodologies presented here. If there are, the authors should emphasize them. Otherwise, this manuscript is very technical, and offers no new insights to the field of seismic microzonation, and its results,

although important to the Town of Avezzano, are very specific.

Specific comments Page 1, line 12, page 4 lines 5, 15, 21 and in many more places – spacing between words is missing. Page 2 line 8: I believe "from" should be "for". Page 2 line 15: "in the hanging wall should be "on the hanging wall". Page 2 lines 16-17: the sentence beginning with "Nevertheless. . ." should be re-written.

———————————————

---

## Referee Comment (RC4) · Anonymous Referee #4 · 30 Nov 2016

GENERAL COMMENTS This paper concerns the results of a basic microzonation study which was carried out in central Italy, in the area of Avezzano. This town was completely destroyed by the 1915 earthquake. According to the Authors, the local geological conditions, due to the location of the town along the marginal areas of an intramontane continental basin, have remarkably contributed to the amplification of the damages. The study of the geological, morphological and structural control-factors on the local seismic site response can thus potentially represent a significant contribution of broad interest, considering that several old villages and towns of central and southern Italy are located in comparable geological and morphological settings. In its present form the manuscript, representing a very good and detailed technical report of

the microzonation studies, has a local relevance and leaves almost unresolved some fundamental issues. Nevertheless, if framed in a more detailed and exhaustive geological model, the robust dataset that includes geological and geophysical data could be a valid base of knowledge to turn the paper in a scientific paper of broad interest. The Authors correctly emphasize the crucial importance of a solid geological model in interpreting the seismic site characterization. Paradoxically, the interpretation of the geological data seems to be too poorly detailed, if compared with the large amount of available information. The individuation of the causes of the local site effects of the 1915 event and the evaluation of future seismic responses on the new town should require a more detailed geological model, beyond the objectives of the basic microzonation analysis, taking into account all the possible aspects (geometry of the thrust system, location and arrangement of the Quaternary normal faults, location of the active and capable faults, lithological lateral variations within the cover units) that may contribute to the amplification of the ground motion. The peculiar complexity of the geological setting of the site imposes the application of 2D or 3D simulation model of the seismic response that requires analyses more advanced than those planned for the basic (first level) seismic microzonation. A possible target of the paper should be to propose a geological model responding to the requirements for a more advanced (third level) seismic microzonation study. The presented data are largely sufficient to achieve this target. In the next specific comments I suggest to the Authors some issues to be solved in the revision of the paper.

SPECIFIC COMMENTS

In the geological setting and in the following sections are almost obscure on both the significance and the role of the SSE-dipping Tre Monti Fault System, during the evolution of the intramontane basin. The Fucino Basin is, in fact, described as an half-graben connected to the motion, since the Late Pleistocene, along the SW-dipping Fucino master fault system. It is unclear the origin of the Lower Pleistocene depocenter (pull-apart? thrust-top?) bordered by the Tre Monti system. This issue is relevant

in term of geometry of the faults along the northern border of the basin that should represent the most impressive tectonic feature buried in the subsurface of Avezzano. The Author should also explain the present-day kinematics of the active segments of the northern border, associated to the normal faulting along the Fucino master normal fault. I would also point out the discrepancies in Fig. 1 where the active faults reported in the map are absent in the geological profile. I suggest to draw an original transect, illustrating the geometry of the active faults and their relation with the previous features of the thrust system. Would be also useful provide a regional scale cross section across the northern border of the basin to help the full understanding of the tectonic setting of the Avezzano area by readers not familiar with the area.

The adopted methodology is inspired to the application of the National Civil Protection guidelines for the first level microzonation studies that are necessary, but clearly insufficient to represent the complexity of the examined geological context. The Authors are thus invited to point out how their approach has significantly improved the guidelines, with specific regards to the implementation of the geological field analyses to employ as tool for deciphering analogous geological setting.

The new structure of the Geological-Technical Map proposed in the paper represent a very good proposal aiming at preserve and better exploit the original geological data on which the entire microzonation process is based. Coherently with this appreciable purpose, I invite the Authors to consider the structural features as part of the base geological map, as they actually are, rather than part of the geomorphological and hydrological map. This would help for a better reconstruction of the 3D subsurface geometry that also include the sharp vertical offsets of the bedrock-covers contact across the main fault planes.

The three types of margin adopted as model do not consider any role, control or indirect interference between the site response and the primary geometry of the Neogene thrust belt of Apennines. Some evidences along the transect G-G', almost parallel to the transect from Cavinato et al., 2002 of Fig. 1, would suggest the possibility of

a different interpretation of the geometry of the margin, consisting of the NE dipping carbonate monocline of Mt. Salvino. The authors extend the monocline beneath the flysch sequences, flooring the Upper Pliocene-Lower Pleistocene alluvial and lacustrine deposits of the basin. They attribute to the flysch strata the same attitude of the carbonates. This implies a sharp strata truncation of the flysch levels along the unconformity at the base of the lacustrine deposits and an exaggerated thickness of the flysch monocline concealed beneath the basin. I invite the Authors to discuss and eventually refuse the geometry of the margin proposed in Cavinato et al. (2002) that interpreted the carbonate monocline as the forelimb of a ramp anticline overthrusting the flysch sequence lying almost parallel to the base of the alluvial and lacustrine deposits.

The geometry of the normal faults buried along the northern border of the basin is too poorly defined if compared with that of the segments exposed at the eastern edge of the same margin. The Authors are invited to better reconstruct the continuity of the buried structures tectonically controlling the margin.

The Authors should try to solve the uncertainties about the lithofacies of Lac 1 and Lac 2 along the marginal areas beneath the town of Avezzano. They are also invited to synthetize the different columns related to the 1D subsurface models referred to the single microzones in a 2D model emphasizing the lateral discontinuities and their possible role in determining the seismic site response.

Finally, the conclusions are too generic. I expect that Authors make some further efforts to define the different causes that were directly responsible for the amplification of the damages in the Avezzano area and consequently delineate the good practises to employ in the geological analyses for the seismic microzonation of similar complex settings.

I hope the Authors can find the above listed comments constructive and useful for the improvement of their paper.

---

## Author Comment (AC1) · 1 Feb 2017

Author reply to referee 1 comments on "Shallow subsurface geology and seismic microzonation in a deep continental basin. The Avezzano town, Fucino basin (central Italy)" by Paolo Boncio et al.

REFEREE: The paper presents the level 1 Seismic Microzonation of the Avezzano town considered as a case study of shallow subsurface and site effects in a deep continental basin environment, focusing on geologic constraints and discussing some methodological procedures.

GENERAL COMMENTS: Despite the topic is interesting and suitable for NHESS (sub-

surface geology and associated local seismic hazard in anthropized continental basin), the goals of the manuscript are not clearly presented; the authors seem to hesitate in identifying the most important and innovative results of their own work, describing data, methods and results (achieved or open questions) without emphasis (e.g. the proposed new structure of Geological-Technical Map for SM is wound up in few sentences without highlighting the strengths and differences with existing guidelines). This is evident from the very beginning: i) the title refers to seismic microzonation in a deep continental basin but data, methods, maps, and results refer almost exclusively to the shallow part of the basin (the Avezzano town and surroundings); ii) the Introduction is too general and vague (e.g. the "additional methodological procedures" proposed as implementation of guidelines for Seismic Microzonation should be stressed and better described due to their relevant applicative implications); iii) in the further paragraphs new data and implementation of new methods are far from being clearly separated from literature or previous data and methods. The paper can be improved by clearly defining the main goals and going through the rewriting of some paragraphs, first of all the Introduction, in order to clarify the perspective. My recommendation is that the manuscript can be accepted after revisions ac-cording to general and specific comments.

RESPONSE: We would like to thank the reviewer for the useful suggestions and we are going to comply all the general comments in the revised version of the manuscript. The queries about the specific comments have been answered separately in the following section.

SPECIFIC COMMENTS

REFEREE:

TITLE: • The title should be modified to better fit with the content of the manuscript. The use of "deep" referred to the (Fucino) continental basin as a whole is misleading and in contrast with "The Avezzano Town", indeed the paper deals with the seismic microzonation of the Avezzano town, that is not the deeper part of the basin. • More

in general, considering the map of fig. 1b, the Fucino basin can't be considered as a deep continental basin due to the very limited extension of its deeper part (Quaternary infill up to 1,000 meters). I suggest to modify the title deleting the catchword "deep"and putting in evidence the case study of the Avezzano town. Please don't use the capital letter for Town.

RESPONSE: We will change the title deleting the word "deep". The new title of the paper is:" Shallow subsurface geology and seismic microzonation of the Avezzano town (Fucino continental basin, central Italy)"

REFEREE:

1. INTRODUCTION: • The section must be rephrased, especially in the first part, where the many clauses don't make a meaningful sentence. • The Authors state that "a geological model is achieved by the interplay of different data, but surface geology obtained by basic detailed geological survey (e.g. 1:5,000 scale), integrated with borehole stratigraphies, still remains the fundamental source of information"; however, considering the morphology of the Fucino basin (flat area) and that the paper focus on the Avezzano town (urbanized area).I suggest the Authors to better explain what they mean and how they have used surface data as fundamental source of information. Also in the paper the subsurface data seem to be the most important ones.

RESPONSE: We accept the comments and we will modify the Introduction. In particular, we will better explain the used geological dataset. It is constituted by the integration of field and subsurface (borehole-derived) data on which the entire microzonation process is based. Moreover, we would like to clear up that an original field geological survey has been performed (1:5000 scale). In spite of the urbanization and relatively flat morphology of the area (please note that the area is flat only where the Lac3 unit crops out), the performed survey helped significantly in constraining the geology of the Avezzano area (e.g., boundaries between cover units and bedrock; boundaries between different units within the continental successions, etc.).

REFEREE:

2. GEOLOGICAL SETTING: • In the text (p. 2 line 25 and p.3 line 5) it is not clear if the Authors consider the Fucino basin as a graben or half-graben; they use (Fig. 1) the geological section from Cavinato et al., 2002 with a half-graben interpretation but the map where the trace of the cross section is indicated reported an articulated graben framework for the Fucino basin not corresponding with the cross section. Please better explain your personal interpretation, based on your own new data, and draw your own geological cross section.

RESPONSE: We agree with this comment. We will draw an original geological transect that takes into account the geometry of the active faults reported in Figure 1b. It is based on our original data for that regarding the western sector, and on the data published by Cavinato et al. (2002) for the central and eastern sectors.

REFEREE:

• The Authors use the term Late Pliocene (p.3 line 1): it would be useful to indicate the chronostratigraphic chart used or to modify the term with Gelasian in order to give a clear and unambiguous chronostratigraphic reference.

RESPONSE: We used the chronostratigraphic chart with the base of the Quaternary placed at 2.588 Ma, thus the term "Late Pliocene" is properly used and updates the definition used in the original paper (Galadini and Messina, 1994). We will specify this point in the revised text.

REFEREE:

3. METHODOLOGY : • The phase 2 "geologic and geomorphologic field survey" is indicated as the "most important during Level 1 SM" and the Authors state "A detailed geological survey has been carried out". However data and results of this activity are not well presented in the paper. As reported in the text, most of the surface geology characteristics are derived or referred to the 1:50,000-scale Italian Geological Map of

the CARG Project (see p. 4 line 33, and pp.6-7). Please try to better explain which are your own new data.

RESPONSE: In the revised text we will clear up that the continental deposits were mapped following an original field geological survey. For that regarding the bedrock units: 1) the field geological survey were performed by the authors in the Tre Monti area, 2) the pre-existing CARG map has been used as reference base map, but it was revised, modified where necessary and adapted to the 1:5,000 scale map for the entire studied area.

REFEREE:

• I suggest to define with short titles the 4 phases and to use the same titles in sections 4-6.

RESPONSE: There is not a perfect correlation between the 4 phases and the sections 4-6. For this reason, we decided to avoid this change. However, in order to improve the text comprehension, we will add within the description of the 4 phases (section 3) short references to the sections where the results from each phase are described: ex. "Phase 1 (see sections 4 and 5)".

REFEREE:

- At the web page linked at page 4 line 33 the geological sheet Avezzano is reported as published in 2005. Please add the correct reference to the map in the text (pp. 4, 6-7) and in the references list, together with the web link.

RESPONSE: We will correct.

REFEREE:

• p. 4 line 22: Fig. 4 is cited before fig. 2 and 3. You can delete this reference, it is not necessary here.

RESPONSE: We will correct.

REFEREE:

• p. 5 line 12 : cover units are distinguished into three classes (as in fig. 2a) and not in two as indicated in the text.

RESPONSE: We will correct.

REFEREE:

4 AND FOLLOWING SECTIONS: • I suggest to use the same titles adopted for the 4 phases described at page 4.

RESPONSE: Please, see the reply to the second comment of the "Methodology" section.

REFEREE:

• Section 5.1 - p. 6 line 28: what do you mean with stratigraphic "domain"?

RESPONSE: Ok, we change "stratigraphic domain" in "stratigraphic succession".

REFEREE:

• Section 5.2. - p. 8 line 28-29, 31: use here the same style for the "name" of the three typologies of shallow subsurface, as in the text at p. 9 lines 16 and 27, and in the figures. i.e.: "A-type....; B-type....; C-type. . . . . .".

RESPONSE: We will modify.

REFEREE:

• p. 9 line 11-14: the time-depth conversion used appears too simple for a basin characterized by high stratigraphic complexity and high lateral variability of the units. Do you have tested different values or more complex velocity models to have a more consistent prediction of the bedrock depth in the whole area?

RESPONSE: We are going to better explain our choices. We would like to point out

that the reconstruction of a more complex velocity model for the area, finalized to a time-depth conversion, unfortunately is not feasible on the base of the available data.

REFEREE:

• Section 6.3 p. 12 line 9: "contained in the MOPS" replace with "summarized in the MOPS".

RESPONSE: We will modify.

FIGURES:

REFEREE:

• Fig. 1 - The isochrones from Cavinato et al., 2002 appear to be modified also outside the rectangle of Fig.4. Please check and/or modify the caption.

RESPONSE: We will modify.

REFEREE:

- The geological cross section should be replaced with your own cross section or the trace moved in fig. 1a. As a matter of fact the trace on the map of fig. 1b creates misunderstanding in the reader due to the differences between faults in the map and in the geological cross section.

RESPONSE: Ok. Please, see also the reply to comment 1 of "Geological setting" section.

REFEREE:

• Fig. 3: not clear/unreadable

RESPONSE: Ok, we will improve the quality of figure 3. However, it is worth to note that the aim of this figure is to show the distribution, types and the amount of pre-existing data.

REFEREE:

• Fig. S1: please indicate that some of the units included in the legend are not represented in the map, or modify the legend.

RESPONSE: We will modify.

REFEREE:

• Fig. S2: please indicate that some of the zones included in the legend are not represented in the map, or modify the legend.

RESPONSE: We will modify.

REFEREE:

REFERENCES • Add reference to the geological sheet Avezzano • Check the correct form of the publication MS Working Group, 2008. In the text is always cited as Working Group SM, 2008. Unify. • SESAME 2004: it is not cited in the text.

RESPONSE: We will correct.

REFEREE:

TECHNICAL CORRECTIONS: • there are several words not separated by blank - p. 13 • line 31: "undergoe" change in "undergo"

RESPONSE: We will correct.

---

## Author Comment (AC2) · 1 Feb 2017

Author reply to referee #2 comments on "Shallow subsurface geology and seismic microzonation in a deep continental basin. The Avezzano town, Fucino basin (central Italy)" by Paolo Boncio et al.

GENERAL COMMENTS

REFEREE:

This paper has only applicative interest because it focuses on the Level 1 SM, which is a routine practice in Italy.

[Figure]

RESPONSE: We would like to thank the reviewer for the comments. Certainly this work has applicative interests. Nevertheless, we think that the scientific community might be interested to this work for a number of points, including:

The structure of the Geological-Technical Map (G-T Map) proposed in this paper represents a new methodological approach compared to that required by the Italian SM guidelines. In fact, the Italian SM guidelines, published in 2008 (see SM Working Group, 2015 * for the English edition), do not provide technical specifications for the G-T Map. Some implementations have been published more recently (e.g., Martini et al., 2011**; "Standard di rappresentazione e archiviazione informatica. Microzonazione sismica. Versione 4.0b, 2015" available online at http://www.protezionecivile.gov.it/resources/cms/documents/StandardMS_4_0b.pdf).
These implementations provide some guidelines for the G-T Map that favor the mapping of textural features for cover soils (gravel, sand, silt, etc.) and geo-mechanical features for the geological bedrock (lapideous vs pelitic vs interlayering, stratification, fracturing, etc.). A number of basic geologic data, necessary for the 3D reconstruction of geological bodies, are lost (chronostratigraphic relations, sedimentary environments, etc.). In any case, specific instructions for building the G-T Map are not provided. The aim of this work is not to modify the Italian guidelines, but we propose an original methodological procedure for building a G-T Map for SM which might be of interest for scientists and professionals working in the field of SM, in Italy or elsewhere. This procedure was adopted for basic (Level 1) SM of the Abruzzo Region. The proposed methodology and the resulting G-T Map preserve basic geological data, and implement them with additional lithological-technical features useful for SM.

This paper represents a new case history in the scientific literature, with potential interest for other areas with similar geologic context; We also agree with the statement from the anonymous Referee #4 "The study of the geological, morphological and structural factors controlling the local seismic site response can potentially represent a significant contribution of broad interest, considering that several old villages and towns of central

and southern Italy are located in comparable geological and morphological settings";

This paper contributes to improve the knowledge of the seismic hazard of this area.

In order to clarify these points, we are going to modify the "Introduction", the "Methodology" and the "Discussion and Conclusion" sections.

The queries about the specific comments have been answered separately in the following section.

SPECIFIC COMMENTS:

REFEREE:

• They write: "25 The Fucino basin is a Quaternary graben" but, strictly speaking, the section of figure 1b does not show a graben. Then, in Fig. 1b the trace of the section intersects 5 red faults, but the section below shows 6.

RESPONSE: We agree with this comment. We will draw an original geological transect that takes into account the geometry of the active faults reported in Figure 1b. It is based on our original data for that regarding the western sector and on the data presented in Cavinato et al. (2002) for the central and eastern sectors.

REFEREE:

• They write that "the SW-dipping Fucino fault system [...] borders the basin towards the east" and "the basin depocenter is localized towards the eastern margin". Rather, it borders the basin towards the NE, and they refer to the north-eastern margin.

RESPONSE: We will correct E with NE.

REFEREE:

• "Pleistocene medium-to coarse-grained fluvial, alluvial fan, and slope-derived deposits are interfingered with lacustrine sediments (Fig. 1b)". Non visible in the figure.

RESPONSE: We agree with this comment; we will delete the reference to figure 1b.

REFEREE:

• "from 1400 and 2600 yrs" from 1400 to 2600 yrs (?). However, 33 kyrs/9 = 3666 yrs. So, "from 1400 to 2600 yrs" does not work. Thus?

RESPONSE: We agree with this comment. We realized that there is a mistake in this sentence. We will modify in "There were 9 earthquakes during the last ∼19 kyrs, including the 1915 event (Serva et al., 1986; Michetti et al., 1996; Galadini and Galli, 1999; Galli et al., 2008, 2012), with an average recurrence time which according to Galadini and Galli (1999) ranges between 1400 and 2600 yrs".

REFEREE:

• I think they used Lennartz 3D/5 sensors. Please mention the used instrumental chain also in this paper, to ensure the reader that detected frequencies < 1Hz are reliable.

RESPONSE: Ok, the information required will be added.

TECHNICAL CORRECTIONS:

REFEREE:

• The Mt. Parasano and Mt. Serrone ridge and the Val Roveto are not shown in the figures.

RESPONSE: Ok, we will add the missing localities in figure 1.

REFEREE:

• Fig. 4 is mentioned before Figs. 2 and 3. Change their order.

RESPONSE: We will correct.

REFEREE:

• "(silt and clay, Fig.1b)" are not shown in Fig. 1b.

RESPONSE: We will delete the reference to figure 1b.

REFEREE:

• Several cases of irregular spacing throughout the text.

RESPONSE: We will correct.

REFEREE:

• The SDTM and the dynamic penetration test, shown in Fig. 3, are not mentioned in the text.

RESPONSE: We will add a mention to the SDTM and the dynamic penetration test in the text.

CITED REFERENCES * SM Working Group (2015) – Guidelines for Seismic Microzonation. Civil Protection Department and Conference of Regions and Autonomous Provinces of Italy. 1 Vol. English edition of: Gruppo di lavoro MS, Indirizzi e criteri per la microzonazione sismica, Conferenza delle Regioni e delle Province autonome – Dipartimento della protezione civile, Roma, 2008, 3 vol. e Dvd. Available online at http://www.protezionecivile.gov.it/httpdocs/cms/attach_extra/GuidelinesForSeismicMicrozonation.pdf ** Martini et al. (2011) in Ingegneria Sismica XXVIII,2, 2011, available online at http://www.protezionecivile.gov.it/resources/cms/documents/aggiornamento_indirizzi_microzonazione_sismica.pdf

---

## Author Comment (AC3) · 1 Feb 2017

Author reply to referee #3 comments on "Shallow subsurface geology and seismic microzonation in a deep continental basin. The Avezzano town, Fucino basin (central Italy)" by Paolo Boncio et al.

REFEREE:

The manuscript describes the construction of a level 1 seismic microzonation map of the Avezzano town in central Italy. It begins with an overview of the studies area, proceeds with describing what a seismic microzonation is, and continuous with describing the types of data collected for this research, and the data itself.

[Figure]

GENERAL COMMENTS:

The work itself is very local, although it can serve as a "guide" for other places in the world. The authors state that the SM was carried out according to the guidelines for the SM by the Italian Department of Civil Protection. It is not understood whether there are new approaches or methodologies presented here. If there are, the authors should emphasize them. Otherwise, this manuscript is very technical, and offers no new insights to the field of seismic microzonation, and its results, although important to the Town of Avezzano, are very specific.

RESPONSE: We would like to thank the reviewer for the comments. We agree that methodological aspects of the work should be better valorized. In fact, the paper has not only local interest as:

The structure of the Geological-Technical Map (G-T Map) proposed in this paper represents a new methodological approach compared to that required by the Italian SM guidelines. In fact, the Italian SM guidelines, published in 2008 (see SM Working Group, 2015 * for the English edition), do not provide technical specifications for the G-T Map. Some implementations have been published more recently (e.g., Martini et al., 2011**; "Standard di rappresentazione e archiviazione informatica. Microzonazione sismica. Versione 4.0b, 2015" available online at http://www.protezionecivile.gov.it/resources/cms/documents/StandardMS_4_0b.pdf). These implementations provide some guidelines for the G-T Map that favour the mapping of textural features for cover soils (gravel, sand, silt, etc.) and geo-mechanical features for the geological bedrock (lapideous vs pelitic vs interlayering, stratification, fracturing, etc.). A number of basic geologic data, necessary for the 3D reconstruction of geological bodies, are lost (chronostratigraphic relations, sedimentary environments, etc.). In any case, specific instructions for building the G-T Map are not provided. The aim of this work is not to modify the Italian guidelines, but we propose an original methodological procedure for building a G-T Map for SM which might be of interest for scientists and professionals working in the field of SM, in Italy or

elsewhere. This procedure was adopted for basic (Level 1) SM of the Abruzzo Region. The proposed methodology and the resulting G-T Map preserve basic geological data, and implement them with additional lithological-technical features useful for SM.

This paper represents a new case history in the scientific literature, with potential interest for other areas with similar geologic context;

We also agree with the statement from the anonymous Referee #4 "The study of the geological, morphological and structural factors controlling the local seismic site response can potentially represent a significant contribution of broad interest, considering that several old villages and towns of central and southern Italy are located in comparable geological and morphological settings";

In order to clarify these points, we will modify the "Introduction", the "Methodology" and the "Discussion and Conclusion" sections.

REFEREE:

SPECIFIC COMMENTS: • Page 1, line 12, page 4 lines 5, 15, 21 and in many more places – spacing between words is missing. • Page 2 line 8: I believe "from" should be "for". • Page 2 line 15: "in the hanging wall should be "on the hanging wall". • Page 2 lines 16-17: the sentence beginning with "Nevertheless. . ." should be re-written.

RESPONSE: We will correct.

CITED REFERENCES * SM Working Group (2015) – Guidelines for Seismic Microzonation. Civil Protection Department and Conference of Regions and Autonomous Provinces of Italy. 1 Vol. English edition of: Gruppo di lavoro MS, Indirizzi e criteri per la microzonazione sismica, Conferenza delle Regioni e delle Province autonome – Dipartimento della protezione civile, Roma, 2008, 3 vol. e Dvd. Available online at http://www.protezionecivile.gov.it/httpdocs/cms/attach_extra/GuidelinesForSeismicMicrozonation.pdf ** Martini et al. (2011) in Ingegneria Sismica XXVIII,2, 2011, available online at

http://www.protezionecivile.gov.it/resources/cms/documents/aggiornamento_indirizzi_microzonazione_sismica.pdf

---

## Author Comment (AC4) · 1 Feb 2017

Author reply to referee #4 comments on "Shallow subsurface geology and seismic microzonation in a deep continental basin. The Avezzano town, Fucino basin (central Italy)" by Paolo Boncio et al.

REFEREE:

GENERAL COMMENTS:

This paper concerns the results of a basic microzonation study which was carried out in central Italy, in the area of Avezzano. This town was completely destroyed by the 1915 earthquake. According to the Authors, the local geological conditions, due to the

location of the town along the marginal areas of an intramontane continental basin, have remarkably contributed to the amplification of the damages.

The study of the geological, morphological and structural control-factors on the local seismic site response can thus potentially represent a significant contribution of broad interest, considering that several old villages and towns of central and southern Italy are located in comparable geological and morphological settings. In its present form the manuscript, representing a very good and detailed technical report of the microzonation studies, has a local relevance and leaves almost unresolved some fundamental issues. Nevertheless, if framed in a more detailed and exhaustive geological model, the robust dataset that includes geological and geophysical data could be a valid base of knowledge to turn the paper in a scientific paper of broad interest. The Authors correctly emphasize the crucial importance of a solid geological model in interpreting the seismic site characterization. Paradoxically, the interpretation of the geological data seems to be too poorly detailed, if compared with the large amount of available information. The individuation of the causes of the local site effects of the 1915 event and the evaluation of future seismic responses on the new town should require a more detailed geological model, beyond the objectives of the basic microzonation analysis, taking into account all the possible aspects (geometry of the thrust system, location and arrangement of the Quaternary normal faults, location of the active and capable faults, lithological lateral variations within the cover units) that may contribute to the amplification of the ground motion. The peculiar complexity of the geological setting of the site imposes the application of 2D or 3D simulation model of the seismic response that requires analyses more advanced than those planned for the basic (first level) seismic microzonation. A possible target of the paper should be to propose a geological model responding to the requirements for a more advanced (third level) seismic microzonation study. The presented data are largely sufficient to achieve this target. In the next specific comments I suggest to the Authors some issues to be solved in the revision of the paper.

RESPONSE: We would like to thank the reviewer for the useful comments. We are going to comply all the general comments. We hope to clear all the requests through our responses below and by revising the text in the manuscript.

The queries about the specific comments have been answered below.

SPECIFIC COMMENTS:

REFEREE:

• In the geological setting and in the following sections are almost obscure on both the significance and the role of the SSE-dipping Tre Monti Fault System, during the evolution of the intramontane basin. The Fucino Basin is, in fact, described as an half-graben connected to the motion, since the Late Pleistocene, along the SW-dipping Fucino master fault system. It is unclear the origin of the Lower Pleistocene depocenter (pull-apart? thrust-top?) bordered by the Tre Monti system. This issue is relevant in term of geometry of the faults along the northern border of the basin that should represent the most impressive tectonic feature buried in the subsurface of Avezzano.

• The Author should also explain the present-day kinematics of the active segments of the northern border, associated to the normal faulting along the Fucino master normal fault.

RESPONSE: We accept the comments and we decided to improve the "Geological setting" section. Also following the comments of Referee #1, we are going to better describe the structural geology of the Fucino basin. We will also clarify some aspects regarding the geometry and kinematics of the Tre-Monti fault system. However, it is worth to note that we mainly refer to available literature, as the present paper deals more specifically with the surface and shallow subsurface geology of the area, not the deep geometry of the fault systems.

REFEREE:

• I would also point out the discrepancies in Fig. 1 where the active faults reported

in the map are absent in the geological profile. I suggest to draw an original transect, illustrating the geometry of the active faults and their relation with the previous features of the thrust system. Would be also useful provide a regional scale cross section across the northern border of the basin to help the full understanding of the tectonic setting of the Avezzano area by readers not familiar with the area.

RESPONSE: We agree with this comment. We will draw an original geological transect that takes into account the geometry of the active faults reported in Figure 1b. It is based on our original data for that regarding the western sector, and on the data published by Cavinato et al. (2002) for the central and eastern sectors.

The relation of the normal fault systems with the thrust system would be an intriguing aspect but it implies the analysis and interpretation of deep geophysical data. In our opinion, this aspect is beyond the focus of this paper.

REFEREE:

• The adopted methodology is inspired to the application of the National Civil Protection guidelines for the first level microzonation studies that are necessary, but clearly insufficient to represent the complexity of the examined geological context. The Authors are thus invited to point out how their approach has significantly improved the guidelines, with specific regards to the implementation of the geological field analyses to employ as tool for deciphering analogous geological setting.

RESPONSE: We accept the comments and we are going to modify the "Introduction", "Methodology" and "Discussion and Conclusion" sections in order to better explain the geological field analysis phase. In particular, we will focus on the structure of the Geological-Technical Map. The structure of the Geological-Technical Map (G-T Map) proposed in this paper represents a new methodological approach compared to that required by the Italian SM guidelines. In fact, the Italian SM guidelines, published in 2008 (see SM Working Group, 2015 * for the English edition), do not provide technical specifications for the G-T Map. Some implementations have been published more

recently (e.g., Martini et al., 2011**; "Standard di rappresentazione e archiviazione informatica. Microzonazione sismica. Versione 4.0b, 2015" available online at http://www.protezionecivile.gov.it/resources/cms/documents/StandardMS_4_0b.pdf). These implementations provide some guidelines for the G-T Map that favour the mapping of textural features for cover soils (gravel, sand, silt, etc.) and geo-mechanical features for the geological bedrock (lapideous vs pelitic vs interlayering, stratification, fracturing, etc.). A number of basic geologic data, necessary for the 3D reconstruction of geological bodies, are lost (chronostratigraphic relations, sedimentary environments, etc.). In any case, specific instructions for building the G-T Map are not provided. The aim of this work is not to modify the Italian guidelines, but we propose an original methodological procedure for building a G-T Map for SM which might be of interest for scientists and professionals working in the field of SM, in Italy or elsewhere. This procedure was adopted for basic (Level 1) SM of the Abruzzo Region. The proposed methodology and the resulting G-T Map preserve basic geological data, and implement them with additional lithological-technical features useful for SM.

REFEREE:

• The new structure of the Geological-Technical Map proposed in the paper represent a very good proposal aiming at preserve and better exploit the original geological data on which the entire microzonation process is based. Coherently with this appreciable purpose, I invite the Authors to consider the structural features as part of the base geological map, as they actually are, rather than part of the geomorphological and hydrological map. This would help for a better reconstruction of the 3D subsurface geometry that also include the sharp vertical offsets of the bedrock-covers contact across the main fault planes.

RESPONSE:  This is a good suggestion. Actually, the structural features are part of a base geological map but, in this context, we prefer to preserve the current layout of the Geological-Technical Map. In fact, layers 1 and 2 are made exclusively of polygon features. Even though not specifically stated in the manuscript, this choice has direct

implications of GIS-aided data process.

REFEREE:

• The three types of margin adopted as model do not consider any role, control or indirect interference between the site response and the primary geometry of the Neogene thrust belt of Apennines.

RESPONSE: The seismic site response is mainly controlled by the presence of high impedance contrast interfaces. In the Fucino basin, it is represented by the contact between the continental basin infill and the bedrock, here represented by carbonate and/or flysch units. For this reason, even if the reconstruction of the primary geometry of the Neogene thrust would be an intriguing aspect, the site response is not significantly influenced by it. This point will be better described during the revision.

REFEREE:

Some evidences along the transect G-G', almost parallel to the transect from Cavinato et al., 2002 of Fig. 1, would suggest the possibility of a different interpretation of the geometry of the margin, consisting of the NE dipping carbonate monocline of Mt. Salvino. The authors extend the monocline beneath the flysch sequences, flooring the Upper Pliocene-Lower Pleistocene alluvial and lacustrine deposits of the basin. They attribute to the flysch strata the same attitude of the carbonates. This implies a sharp strata truncation of the flysch levels along the unconformity at the base of the lacustrine deposits and an exaggerated thickness of the flysch monocline concealed beneath the basin. I invite the Authors to discuss and eventually refuse the geometry of the margin proposed in Cavinato et al. (2002) that interpreted the carbonate monocline as the forelimb of a ramp anticline overthrusting the flysch sequence lying almost parallel to the base of the alluvial and lacustrine deposits.

RESPONSE: Actually, our reconstruction and that proposed by Cavinato et al. (2002) are not so different. Section G-G' is only a small portion of the section proposed by

Cavinato et al. (westernmost portion of the section). In any case, in the revised version we will discuss the similarities/differences with existing literature.

REFEREE:

• The geometry of the normal faults buried along the northern border of the basin is too poorly defined if compared with that of the segments exposed at the eastern edge of the same margin. The Authors are invited to better reconstruct the continuity of the buried structures tectonically controlling the margin.

• The Authors should try to solve the uncertainties about the lithofacies of Lac 1 and Lac 2 along the marginal areas beneath the town of Avezzano.

RESPONSE: We accept the comments. We will add a further geological section (H-H', see Figure 7 for the location) in which we provide more details on the geometry of the normal faults and on the contact between Lac1 and Lac2.

REFEREE:

• They are also invited to synthesize the different columns related to the 1D subsurface models referred to the single microzones in a 2D model emphasizing the lateral discontinuities and their possible role in determining the seismic site response.

RESPONSE: It is a good suggestion. We would like to produce a new figure in which we provide a 2-D synthesis of the 3 types of margin, using the MOPS simplified stratigraphic logs.

REFEREE:

• Finally, the conclusions are too generic. I expect that Authors make some further efforts to define the different causes that were directly responsible for the amplification of the damages in the Avezzano area and consequently delineate the good practices to employ in the geological analyses for the seismic microzonation of similar complex settings.

RESPONSE: Following also the suggestions by Referee #1, in the revised version of the manuscript we will modify the "Introduction", "Methodology" and "Discussion and Conclusion" sections in order to highlight the main results, in terms of both methodology (Geological-Technical Map) and implications for geological factors controlling site response in the Avezzano town.

REFEREE:

I hope the Authors can find the above listed comments constructive and useful for the improvement of their paper.

RESPONSE: We thank the Referee, who made really constructive comments.

CITED REFERENCES * SM Working Group (2015) – Guidelines for Seismic Microzonation. Civil Protection Department and Conference of Regions and Autonomous Provinces of Italy. 1 Vol. English edition of: Gruppo di lavoro MS, Indirizzi e criteri per la microzonazione sismica, Conferenza delle Regioni e delle Province autonome – Dipartimento della protezione civile, Roma, 2008, 3 vol. e Dvd. Available online at http://www.protezionecivile.gov.it/httpdocs/cms/attach_extra/GuidelinesForSeismicMicrozonation.pdf ** Martini et al. (2011) in Ingegneria Sismica XXVIII,2, 2011, available online at http://www.protezionecivile.gov.it/resources/cms/documents/aggiornamento_indirizzi_microzonazione_sismica.pdf
* * *